# Investigation on Water Vapor Adsorption of Silica-Phosphonium Ionic Liquids Hybrid Material

**DOI:** 10.3390/ma12111782

**Published:** 2019-06-01

**Authors:** Cancan Li, Jiamei Zhu, Min Zhou, Shuangquan Zhang, Xiaodong He

**Affiliations:** School of Chemical Engineering and Technology, China University of Mining & Technology, Xuzhou 221116, China; 17851145012@163.com (C.L.); zmcumt@126.com (M.Z.); cumtzsq@126.com (S.Z.); researchcumt2017@126.com (X.H.)

**Keywords:** ionic liquids, water vapor, modified silica, adsorption

## Abstract

Adsorption and diffusion of water vapor in phosphonium ionic liquid modified silica gel were studied, aiming to reduce the loading of water vapor in porous materials. The modified silica gel was prepared through a grafting method and characterized by FTIR, thermal gravity analysis and X-ray photoelectron spectroscopy. N_2_ sorption isotherms at −196 °C and CO_2_ sorption isotherms at 0 °C were also measured to analyzee the porosity. Water vapor adsorption equilibriums at 25 °C up to 30 mbar were tested. The results indicate that the ionic liquids (ILs) phase acts as a protecting film which decreases water vapor adsorption. The improvement of water-resistant performance is also attributed to the decrease of micro-porosity and silanol groups on the silica surface. Diffusion behavior of water vapor on modified silica was determined on the basis of the adsorption equilibrium. The effective diffusivity of water vapor in modified silica is almost the same as in bare silica and decreases with the increasing of water vapor loading.

## 1. Introduction

Ionic liquids (ILs) are salts with a melting point below 100 °C, which are commonly composed of organic cations and inorganic or organic anions, typically. ILs have been considered as promising solvents in separation [1,2,3], chemical reaction, or catalytics, due to their unique properties, such as low volatility, chemical stability, high gas selectivity, and tunable properties [4,5]. A lot of research into imidazolium-based ILs have been reported [6,7,8], while phosphonium-based ILs are currently appear more attractive, owing to their unique advantages in many aspects. Phosphonium salts are generally more thermally stable and have lower densities than nitrogen-based ILs [9,10]. Moreover, compared with imidazolium and pyridinium ILs, phosphonium ILs have no acidic proton or aromatic ring, which makes them stable towards nucleophilic and basic conditions [11,12]. In addition, the kinetics of phosphonium salt formations are much faster than those of nitrogen-based salts, implying higher productivity and less expense than their equivalent imidazolium ILs [13,14,15]. Based on the characteristics of ILs, there have been many studies on IL-modified porous materials, such as activated silica, carbon, and alumina for CO_2_ adsorption. Zhang [16] showed that CO_2_ capture capacity increased markedly for grafting amino-functionalized phosphonium ILs ([aP_4443_]AA) onto silica gel than in a bulk liquid and bare support. Balsamo [17] reported that supported [Emim][Gly] ionic liquid on mesoporous gamma-Al_2_O_3_ at different loadings had better CO_2_ sorption capacity. These experiments reveal that CO_2_ capture performance is greatly enhanced owing to the fact that ILs were supported on the porous support.

A noticeable influence of water on the physical characteristics of adsorbents has been also reported, due to the presence of a small amount of water vapor (8–15%) in the power plant flue gas, which tends to significantly reduce CO_2_ adsorption capacity [18,19,20]. Because of the competition adsorption of CO_2_ and H_2_O, the adsorbed water occupies a part of adsorption sites, which is unfavorable for CO_2_ adsorption. Li [21] studied that CO_2_ adsorption of activated alumina strongly decreased with increasing water content in the vapor-gas. Xu [22] investigated the effect of water on CO_2_ adsorption properties in a commercial activated carbon, and cyclic adsorption experiments with wet feed gas showed that the CO_2_ adsorption capacity of activated carbon was negatively affected by the presence of water. Uehara [23] studied CO_2_ sorption behaviors of supported amino acid ionic liquid [EMIM]AA sorbents on poly(methyl methacrylate) (PMMA) under a humidified gas flow condition (2.3% CO_2_, 2.1% H_2_O and balance N_2_), and the results showed that CO_2_ sorption capacities of [EMIM][Gly]- and [EMIM][Lys]-PMMA decreased in the presence of water vapor as the adsorbed water inhibited their reaction with CO_2_.

Meanwhile, the water vapor adsorption performance of the pores adsorbent, especially silica gel, has been investigated extensively. Alcañiz-Monge [24] reported that water adsorption at low relative pressures occurred mainly in the micro-porosity of silica. Saliba [25] also showed that the adsorption of water vapor on porous silica depended upon the concentration of surface hydroxyl groups (silanol) and the porosity. At present, it has been reported that the water vapor adsorption capacity was affected by the pore size and surface hydroxyl groups, but little research has been done to evaluate the surface properties and pore structure of silica gel modified by ILs and different pretreated methods for the supporter, which shows further influence of water vapor adsorption and its water vapor adsorption performance. Thus, we considered surface-modifying and regulating the pore size of silica to be an effective way of controlling the amount of adsorbed water vapor in competitive adsorption.

The water hold-up needs to be addressed in adsorption processes under humid conditions, as this can reduce the adsorption capacity for CO_2_. Based on this starting point, in this work, phosphonium ILs, especially hydrophobic ionic liquids, were chosen to be supported on pretreated silica to prevent the water vapor entering effectively. According to the literature [18,19,20,21,22,23], the surface performance and porosity of porous materials are main effects for competitive adsorption of H_2_O and CO_2_. We assume that pore structure and surface hydroxyl content of silica gels functionalized with ILs will be changed, which can effectively play the advantages to reduce water vapor sorption capacity of porous materials, aiming to prepare a new adsorbent with better solid supported ILs which achieve better water-resistance performance for being more suitable for post-combustion CO_2_ capture applications. Simultaneously, the diffusion of water vapor in IL-modified silica was investigated in order to characterize the influence of the carrier surface and porosity.

## 2. Materials and Methods

### 2.1. Materials

Tri-n-octylphosphine (purity: 97%) was purchased from Stream Chemicals, Inc. (Cambridge, UK). Bistrifluoromethanesulfonimide lithium (purity: 99%) was purchased from J&K Scientific (Beijing, China). 3-Chloropropyltriethoxysilane (purity: 97%) was purchased from Beijing Fengtuo Chem Co., Ltd., Beijing, China. Toluene (dried by Na), acetone, and HCl were all analytical grade. All chemicals were purchased from a commercial supplier and used as received. Silica gel with 0.5–1.5 mm in particle size was purchased from Shanghai Silica Gel Factory, Shanghai, China. Silica was pretreated before use.

### 2.2. Synthesis of grafted silica gel

The pretreated methods of silica affect the pore structure and hydroxyls of the silica surface, thus influencing water vapor sorption capacity. In this study, two different treatement methods were used. Silica was calcined at 150 °C for 3.5 h (coded as SiO_2_-C) or acid treated with hydrochloric acid (8 mol/g) at 70 °C for 12 h (coded as SiO_2_-H). It was then washed with distilled water until neural, and dried at 353 K under vacuum for 8 h before further use. The pretreated silica is referred to as bare silica in the following text.

The preparation of phosphonium IL-modified silica using grafted method was carried out, according to a previous report [26]. In this work, another kind of silica with different pretreated methods was selected for further controlled amounts of IL. The general procedure is as follows. Tri-n-octylphosphine was added to 3-chloropropyltriethoxysilane and anhydrous toluene and stirred under nitrogen gas at 110 °C for 40 h. Phosphonium chloride was then stirred with an aqueous solution of bistrifluoromethanesulfonimide lithium at room temperature for 30 h. The resulting crude IL was extracted from the aqueous phase using dichloromethane. The dichloromethane phase was then washed with distilled water for several times until no residual chloride salt was detected with the use of AgNO_3_. The IL was dried in vacuum at 333 K for 8 h. The pretreated silica was added to the prepared grafting agent in anhydrous toluene under a nitrogen atmosphere and stirred at 90 °C for 24 h. The solid was collected and washed with acetonitrile and distilled water and then dried at 80 °C overnight under vacuum to obtain an IL-modified silica. The product prepared by calcined silica was labeled as Si-P_8_TFSI/SiO_2_-C, when the molar ration of silica and IL is 1:1. The product prepared by acid-treatment silica was coded as Si-P_8_TFSI/SiO_2_-H(I) and Si-P_8_TFSI/SiO_2_-H(II), respectively, depending on the different molar ratio of silica and the grafting agent (1:0.5 and 1:1), as illustrated in Figure 1.

### 2.3. Characterization

#### 2.3.1. Chemical Analysis

Fourier transform infrared (FTIR) spectra were recorded on a Thermo Scientific Nicolet 380 IR Spectrometer (Thermo Scientific, Massachusetts, USA), and the samples, in powder form, along with KBr, were compressed into tablets. The spectra (32 scans for the pure ionic liquid and 128 scans for the IL-modified silica) were collected with a spectral resolution of 4 cm^−1^.

An x-ray photoelectron spectroscopy measurement was carried out on an X-ray Photoelectron Spectrometer ESCALAB 250Xi (Thermo Fisher Co., Franklin, MA, USA). All binding energies were referenced to the neutral C1 speak at 284.8 eV to compensate for the surface-charging effects.

Thermogravimetric analysis (TGA) was performed at a heating rate of 10 °C/min in the range from room temperature to 700 °C under nitrogen atmosphere, using a STA409C DTA/DSC-TG (Netzsch, Selb, Germany). The hydroxyl group contents were estimated by assuming that the condensation occurred based on the reaction of two hydroxyl groups on the silica surface and can be calculated as follows in Equation (1) [27]. Moreover, according to the thermogravimetric curve, the total IL load on the surface of silica gel is determined by using Equation (2).
(1)$$n_{\mathrm{OH}}\left({\mathrm{SiO}}_{2}\right)=\frac{2[W\left(T_{0}\right)-W(T_{\mathrm{final}})]}{100M_{H_{2}O}}$$
(2)$$\mathrm{GR}\left(\%\right)=\frac{W\left(T_{0}\right)-W(T_{\mathrm{final}})}{W(T_{\mathrm{final}})}\times 100$$
where $n_{\mathrm{OH}}\left({\mathrm{SiO}}_{2}\right)$ indicates the moles of hydroxyl groups on the silica particles $W\left(T_{0}\right)$ and $W\left(T_{\mathrm{final}}\right)$ are the weight of the silica particles (g) at the temperatures 200 °C and 700 °C, respectively, M_H2O_ is the molecular weight of water and GR (wt%) is the total load of ILs on the surface of silica gel.

#### 2.3.2. Pore Structure Characteristics

The adsorption isotherm of N_2_ at −196 °C and CO_2_ at 0 °C were measured by an Autosorb-1-MP (Quantachrome, Florida, USA), an automatic surface area and pore size analyzer. The porous particles were dried and degassed at 100 °C under high vacuum for 5 h to remove moisture or other volatile contaminants. As for N_2_ adsorption, the relative pressure (P/P_0_) range was set between 0.001 and 0.995, and the CO_2_ adsorption isotherms were collected within the relative pressure (P/P_0_) range of 0.0001~0.03 for restricting pores narrower than 2 nm. A density functional theory (DFT) analysis was used to calculate the pore size distribution (PSD) and the pore structure parameters using Quantachrome ASiQwin software (Quantachrome, Florida, USA). based on the adsorption isotherms.

### 2.4. Adsorption Studies

Adsorption and diffusivity of water vapor were measured on an Intelligent Gravimetric Analyzer (IGA-003, Hiden Isochema Ltd., Manchester, UK). Detailed information on the apparatus and analyzing methods have been previously reported [28,29]. Water vapor uptake was measured in the Analyzer’s Humidifier configuration, using similar methods at 25 °C. The samples were dried by circulating dry nitrogen until a constant weight was obtained. Vapor pressure was then controlled by circulating a mixture of dry nitrogen and vapor-saturated nitrogen to reach the required humidity. At a required humidity, the mass gain was measured as a function of time until reaching constant. More details have been reported in references [30,31].

## 3. Results and Discussion

### 3.1. Characterization of the Samples

#### 3.1.1. FTIR Analysis

The FTIR spectra of bare and grafted silica are shown in Figure 2. The typical peaks for SiO_2_-H and SiO_2_-C were found at 1098 cm^−1^ for asymmetric stretching vibration and 800 cm^−1^ for symmetric stretching vibration of Si-O-Si. The vibration bands of OH groups are located at different frequencies, which depend on the bonding configuration. The broad band between 3800 cm^−1^ and 3050 cm^−1^ is attributed to the stretching of silanol associated with surface silanols, including isolated silanol, vicinal, and geminal groups [32]. The shoulder at 956 cm^−1^ indicates the existence of the Si-OH asymmetric stretching band [33]. Based on the comparison of the intensity of the characteristic peaks of Si-OH in Figure 2a,b, Figure 2 reviews that hydroxyl content on the surface of SiO_2_-H treated by hydrochloric acid and SiO_2_-C treated by calcined are a little different. The hydroxyl content was calculated in the scetion of thermogravimetry analysis.

As shown in Figure 2, the infrared spectrum of Si-P_8_TFSI/SiO_2_-H(I) shows the appearance of the bands at 2924 cm^−1^ and 2856 cm^−1^ due to C-H aliphatic stretching vibration, and the peak at 1465 cm^−1^ belongs to P-C stretching vibration. Figure 2 appears to show that broad peak characteristic bands between 1024~1246 cm^−1^, due to Si-O-Si and S=O, and the peak at 664 cm^−1^ belongs to S-C stretching vibration. The success of the neat phosphonium IL grafted onto the silica surface has been assessed based on the appearance of the characteristic bands for neat IL at 2926, 2855, 1466, and 1227~1060,663 cm^−1^.

#### 3.1.2. XPS Analysis

The XPS spectrum of SiO_2_-H, neat [Si-P_3888_][TFSI], and Si-P_8_TFSI/ SiO_2_-H(I) is shown in Figure 3. The peaks of C 1s, O 1s, and Si 2p appear only in spectrum (a) of the bare silica. It can be seen from Figure 3b that a binding energy of 168.4 eV, 397.4 eV, and 688.9 eV is attributed to S 2p, N 1s, and F 1s for the neat IL, respectively, in [TFSI] anion [34]. In addition, there is a P 2p peak with a binding energy of 131.0 eV, which further proves the successful synthesis of the ionic liquid [Si-P_3888_][TFSI]. Compared to spectrum (a), the C 1s peak in spectrum (c) is significantly enhanced, and two new peaks attributable to F 1s (binding energy 689.0 eV) and P 2p (binding energy 131.1 eV) appear in the XPS spectrum. This shows that the ionic liquid has been successfully immobilized on the surface of silica.

#### 3.1.3. Thermogravimetric Analysis

Thermal gravimetric analysis (TGA) is performed to determine the quantitative information of the contents of silanol and ionic liquids on the surface of the porous silica products. TGA patterns of bare silica and modified silicas are depicted in Figure 4. It is found that there are two distinct mass-loss steps in the thermogravimetric curves of all samples, especially the abrupt change of the first step at temperatures below 200 °C for SiO_2_-H and SiO_2_-C. The removal of the physiosorbed surface adsorbed water and gas are lost before 200 °C. The condensation of the hydroxyl group and the loaded ionic liquid on the silica surface are lost at the temperature ranging from 200 °C to 700 °C.

Silanol group content for silica and the total load of phosphonium ILs on the surface for modified samples determined by TGA techniques are listed in Table 1. The condensation of all kinds of silanol groups occurs by the reaction of two silanol groups on the silica surface, resulting in the release of one molecule of water and the formation of one siloxane group. It is assumed that no other groups than water were released from the sample. As shown in Table 1, for SiO_2_-H and SiO_2_-C, TGA shows that the hydroxyl content of SiO_2_-C is significantly higher than that of SiO_2_-H, which is probably because the H^+^ in the hydrochloric acid activating solution destroys the hydroxyl groups on the surface of the silica gel [35]. Moreover, IL loading calculated for the Si-P_8_TFSI/SiO_2_-H(I), Si-P_8_TFSI/SiO_2_-H(II), and Si-P_8_TFSI/SiO_2_-C was 4.92 wt%, 10.62 wt%, and 10.79 wt%, respectively, whereby the ionic liquid loading depends on the limited number of active silanol groups on the supporter surface.

#### 3.1.4. Pore size distribution

There is a certain correlation between pore structure and water adsorption. The use of both adsorbates, N_2_ and CO_2_, provides complementary information about the porous texture of the samples. Isotherms measured with N_2_ often lead to an underestimation of the microporosity due to the lower mobility of N_2_ at −196 °C and whose diffusion in small size pores is inhibited. To access the microporosity, isotherms can be measured under CO_2_ at 0 °C, since the CO_2_ molecules are more mobile. Based on the N_2_ and CO_2_ adsorption isotherms at −196 °C and 0 °C, respectively, CO_2_ adsorption capacity at 0 °C and the data to evaluate their surface areas and pore structures of the samples were obtained using the DFT method [36] in Table 2. The DFT model is based on the molecular statistical thermodynamics equation that calculates the specific adsorption amount in an individual pore of a given adsorbate–adsorbent system at a given experimental temperature and pressure by solving the function of the grand thermodynamic potential in terms of the distribution of gas density in a specific pore space.

The cumulative pore volume (CPV) and cumulative surface area (CSA) of SiO_2_-C shown in Table 2 are considerably higher than those of SiO_2_-H, and the surface modified silica presents a significantly lower pore volume and specific surface area as compared to the bare silica, which leads to a decrease in the adsorption of CO_2_. On the basis of structural characteristics, silica impregnated with ILs results in the blockage of pores. Not surprisingly, Si-P_8_TFSI/SiO_2_-C prepared by grafting ILs on calcined supporter has a higher pore volume and specific surface area than Si-P_8_TFSI/SiO_2_-H(II).

The pore size distribution based on N_2_ sorption isotherms at −196 °C and CO_2_ sorption isotherms at 0 °C are shown in Figure 5. According to the pore size distribution (PSD) of the micropore in Figure 5b, the PSD of samples mainly concentrated in the range of 0.4–1.0 nm. The proportion of the micropores for Si-P_8_TFSI/SiO_2_-H(II) is lower than that of Si-P_8_TFSI/SiO_2_-H(I) and Si-P_8_TFSI/SiO_2_-C, which is mainly due to the clogging of a part of the pores after the introduction of the ionic liquid. Combined with IL loading obtained by TGA data, it is expected that the Si-P_8_TFSI/SiO_2_-H(II) with higher IL loading and less micropores should exhibit lower water adsorption capacity, which will be discussed later.

### 3.2. Water Vapor Adsorption

The mechanism of water adsorption on silica is very complex. It is known that the surface of silica contains silanol or siloxane groups. The Si-OH shows a specific interaction with water (hydrophilic), and it is a typical polar group and has some similar properties with water, so the hydrogen bonds were formed by the reaction of water and the hydroxyl group [37,38]. It is reasonable to think that water vapor adsorption, depending on the amount of the hydroxyl group on the silica surface and the adsorption capacity, increased with the hydroxyl content. Water vapor adsorption isotherms, corresponding to all samples, are illustrated in Figure 6. SiO_2_-H has better water resistance than SiO_2_-C. SiO_2_-H displays a lower quantity of the hydroxyl group, as demonstrated by the FTIR and TGA data. According to the pore structure parameters, SiO_2_-H also presents lower CSA and less micropores, which is unfavorable for the adsorption of water vapor.

The results in Figure 6 also indicate that water adsorption of silica modified by ionic liquid reduces significantly, and Si-P_8_TFSI/SiO_2_-H(II) with relatively higher IL content and a lower microporous structure shows better water resistance. Due to grafting the hydrophobic ionic liquid onto the silica gel, there is an ionic liquid film formed by reacting with the silanol groups on the silica surface, and the schematic is shown in Figure 7. According to the previous PSD data, the CPV, especially of the micropores, and CSA of Si-P_8_TFSI/SiO_2_-H(II) are lower, resulting in a slight decrease in water adsorption capacity.

The pore structure parameters and surface property show a correlation with water vapor sorption capacity, suggesting that porosity and the IL film play an important role in reducing a small amount of water vaporin adsorption of the power plant flue gas for the bare and modified SiO_2_, thereby resulting in decreased CO_2_ adsorption capacity. The maximum adsorption capacity of water vapor on silica-phosphonium ionic liquid hybrid material is lower than that for bare silica, with a higher pore volume and quantity of the hydroxyl group, which is advantageous, as less water vapor will be coadsorbed with CO_2_ at the high relative humidities than can be expected in post-combustion applications.

### 3.3. Water Diffusion Coefficient

The diffusion coefficient calculation presented a great significance toward the evaluation of the industrial application value of the adsorbent. The diffusion in porous solids is more complicated than in liquids. For bare silica and silica-phosphonium ionic liquid hybrid material prepared with the grafting method, the particle is considered as a uniform adsorbent. Diffusion of water vapor into a porous spherical particle can be described by Equation (3) [39].
(3)$$\left(1-\varepsilon \right)\frac{\partial q}{\partial t}+\varepsilon \frac{\partial C}{\partial t}=\varepsilon D_{e}\left(\frac{\partial^{2}C}{\partial R^{2}}+\frac{2}{R}\frac{\partial C}{\partial R}\right)$$
where *D_e_* is the effective porous diffusion coefficient (m^2^/s), *ε* is the material porosity, q is the adsorbate concentration in the solid phase (wt%), *C* is the adsorbate concentration in the gas phase (wt%), *R* is the particle of radius (m), and *t* is the adsorption time(s).

Initial conditions (IC) and boundary conditions (BC):
(4)$$C\left(R_{p},0\right)=C_{0}\;q\left(R_{P},0\right)=q_{0}$$
(5)$$C\left(R_{p},\infty \right)=C_{\infty }\;q\left(R_{P},\infty \right)=q_{\infty }$$
$${\frac{\partial C}{\partial R}|}_{R=0}={\frac{\partial q}{\partial R}|}_{R=0}=0$$


The solution of the equation is given by Equation (6) [40].
(6)$$\begin{array}{cccccc} \frac{m}{m_{\infty }}=\frac{q-q_{0}}{q_{\infty }-q_{0}} & & & & = & 1-\frac{6}{\pi^{2}}\sum_{n=1}^{\infty }\frac{1}{n^{2}}exp(\frac{-n^{2}\pi^{2}\left(\varepsilon D_{e}/\left(\varepsilon +\left(1-\varepsilon \right)K\right)\right)t}{R_{p}^{2}})\\ & & & & & =1-\frac{6}{\pi^{2}}\sum_{n=1}^{\infty }\frac{1}{n^{2}}exp(\frac{-n^{2}\pi^{2}D_{ap}t}{R_{p}^{2}}) \end{array}$$
where
(7)$$D_{ap}=\varepsilon D_{e}/\left[\varepsilon +\left(1-\varepsilon \right)K\right]$$
*D_ap_* is the apparent diffusion coefficient, *q* is the quantity adsorbed at time *t*, and *K* is the slope of adsorption isotherm.

A simplified equation for the diffusion is given by Equation (8):
(8)$$\frac{m}{m_{\infty }}\approx \frac{2A}{V}\sqrt{\frac{D_{ap}t}{\pi }}=S\sqrt{t}$$
where *A* is the particle surface area (m^2^) and *V* is the particle volume (m^3^). In case of spherical particles: 2*A*/*V* = 3/*R*.

From equation (8) it can be obtained that:
(9)$$S=\frac{6}{\sqrt{\pi }}\sqrt{\frac{D_{ap}}{R_{p}^{2}}}$$
where *S* is the slope of the curve *m*/*m*_∞_ versus t^0.5^. Further, from Equation (9), *D_ap_* can be obtained as:
(10)$$D_{ap}=\frac{\pi }{36}R_{p}^{2}S^{2}$$


Finally, the effective diffusion coefficient (*D_e_*) can be calculated with the relationship:
(11)$$D_{e}=D_{ap}\frac{\varepsilon +\left(1-\varepsilon \right)K}{\varepsilon }$$
where,
(12)$$\varepsilon =\frac{V_{p}\times \rho_{si}}{1+V_{p}\times \rho_{si}}$$
*V_p_* is the pore volume and *ρ_si_* is the apparent density.

Examples of the experimental data and fitting curves for SiO_2_-H, SiO_2_-C, and Si-P_8_TFSI/SiO_2_-H(I) at different pressures and at 25 °C are illustrated in Figure 8. A good agreement between the experimental values and the fitting curves was obtained.

Diffusion coefficients obtained from the curve fitting are listed in Table 3. It can be obtained from Table 3 that the calculated effect diffusion coefficients of silica gel are at a level of 10^−7^ and these values are close to reports on silica [41,42,43]. Comparing two kinds of bare silica, the diffusion coefficient of SiO_2_-C is lower than that of SiO_2_-H. The reason may be that the diffusion resistance of water vapor is increased by an excessive amount of water adsorbed by SiO_2_-C with a higher hydroxyl group, CPV, and CSA. Moreover, diffusion coefficients in Si-P_8_TFSI/SiO_2_-H(I) are about 10^−8^~10^−7^ m^2^/s and slightly lower than those of SiO_2_-H, which implies that the thin IL coating on the silica surface and a loss of micro-porosity impedes water diffusion.

Figure 9 shows the relationship between water vapor uptake and D_e_. Effective diffusion coefficients of all the samples show a decreasing trend with the increase of water vapor adsorption, which, due to the porosity available for water vapor transport, decreases at higher loading as adsorbed water vapor occupies a part of porous sites.

## 4. Conclusions

Phosphonium-based ILs were successfully immobilized on bare silica pretreated by hydrochloric acid or calcination. Adsorption and diffusion of water vapor on bare and modified silica were investigated. It can be seen from the analyses of FTIR, TG, and PSD that SiO_2_-H has less hydroxyl groups and micropore volumes, which is unfavorable for the adsorption of water vapor, thus it has better water resistance. After the grafting of ILs on silica, the sorbents form a hydrophobic film on the surface to block the entry of water molecules and retain partial pore characteristics of the support. Therefore, Si-P_8_TFSI/SiO_2_-H(II), with relatively higher IL loading and lower micropore, increases the water-resistance performance. Fitting the model with experimental data is beneficial and suggests that the effective diffusion coefficients of bare silica are at a level of 10^−7^, while SiO_2_-H, with relatively rapid diffusion, as lower content of water vapor is benefit for diffusivity. The diffusion of immobilized ionic liquids on silica did not change much and most of the values under different water vapor pressure were also at a level of 10^−7^, while a slight reduction of the diffusivity was observed due to the partial impairing of the pore structure and the forming of a hydrophobic ionic liquid film on the surface. Furthermore, with the increase of water vapor loading, effective diffusion coefficients give a decreasing trend.

## Figures and Tables

**Figure 1 materials-12-01782-f001:**
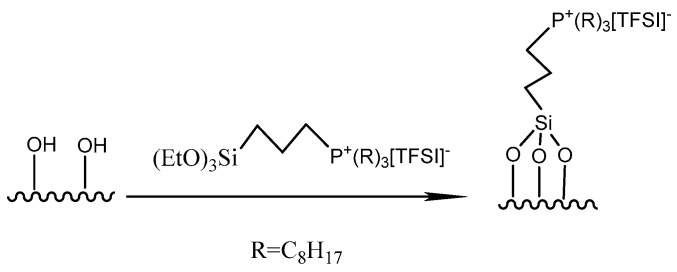
Illustration of synthesizing (EtO)_3_Si[P_8883_]TFSI ionic liquid-modified silica.

**Figure 2 materials-12-01782-f002:**
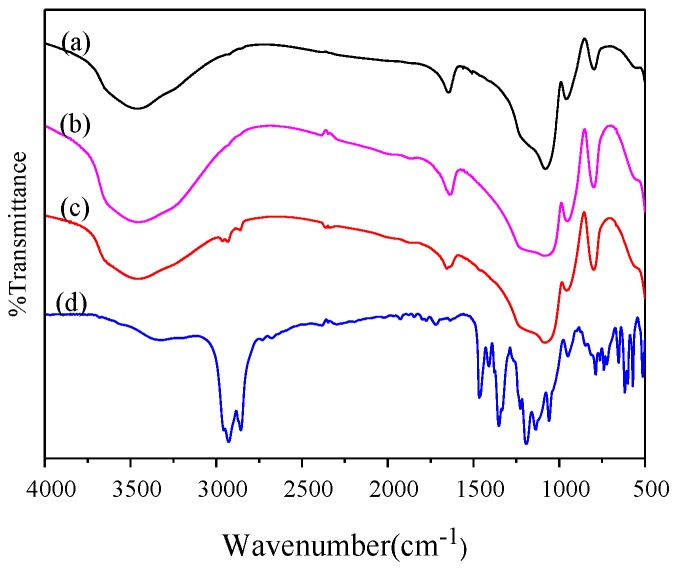
FTIR spectra of SiO_2_-H (**a**), SiO_2_-C (**b**), Si-P_8_TFSI/SiO_2_-H(I) (**c**), and neat ionic liquid (IL) (**d**).

**Figure 3 materials-12-01782-f003:**
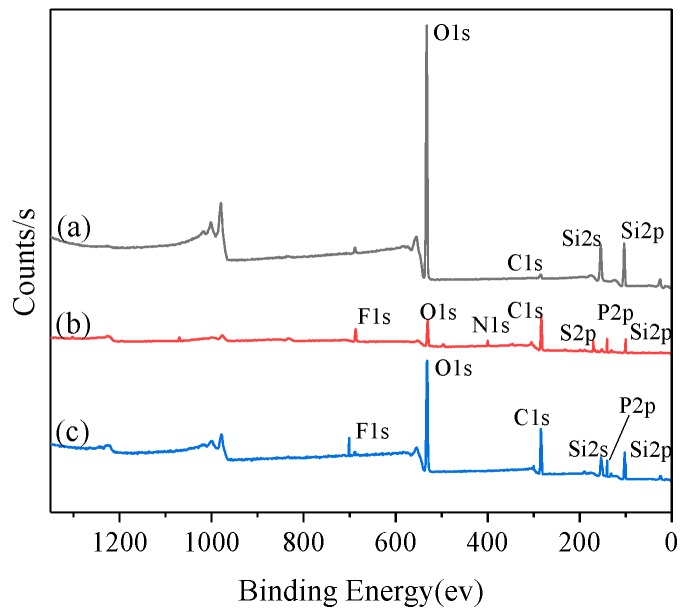
Full-range X-ray photoelectron spectroscopy spectra of SiO_2_-H (**a**), neat IL (**b**), and Si-P_8_TFSI/ SiO_2_-H(I) (**c**).

**Figure 4 materials-12-01782-f004:**
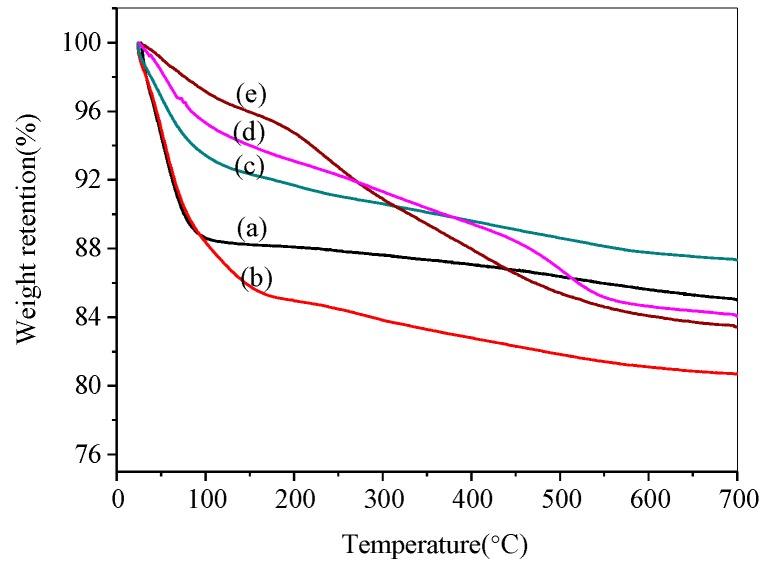
Thermogravimetric curves of SiO_2_-H (**a**), SiO_2_-C (**b**), Si-P_8_TFSI/SiO_2_-H(I) (**c**), Si-P_8_TFSI/SiO_2_-H(II) (**d**), and Si-P_8_TFSI/SiO_2_-C (**e**).

**Figure 5 materials-12-01782-f005:**
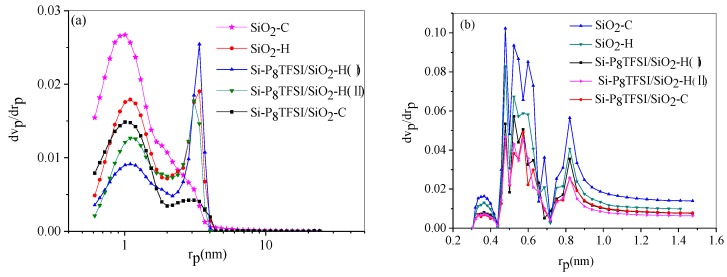
Pore size distribution of silica and modified silica based on: (**a**) N_2_ adsorption isotherms at −196 °C and (**b**) CO_2_ adsorption isotherms at 0 °C.

**Figure 6 materials-12-01782-f006:**
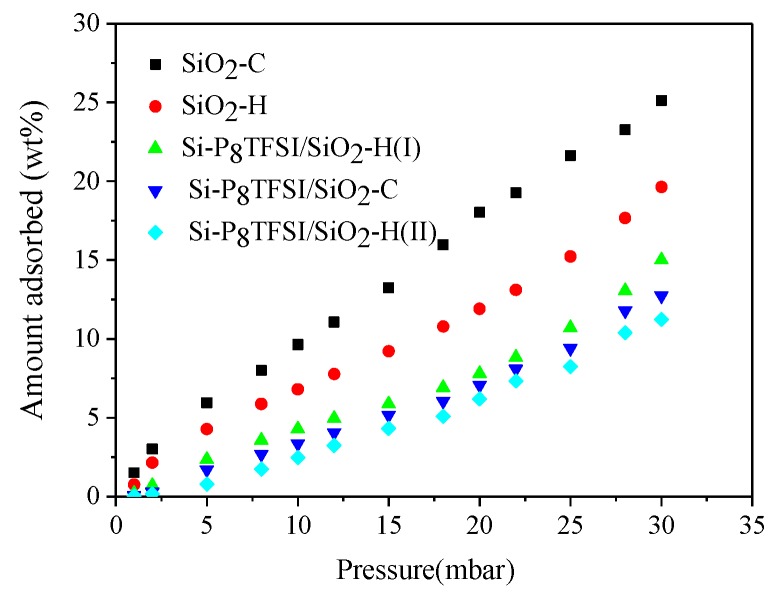
Water vapor adsorption isotherms at 25 °C on silica and modified silica.

**Figure 7 materials-12-01782-f007:**
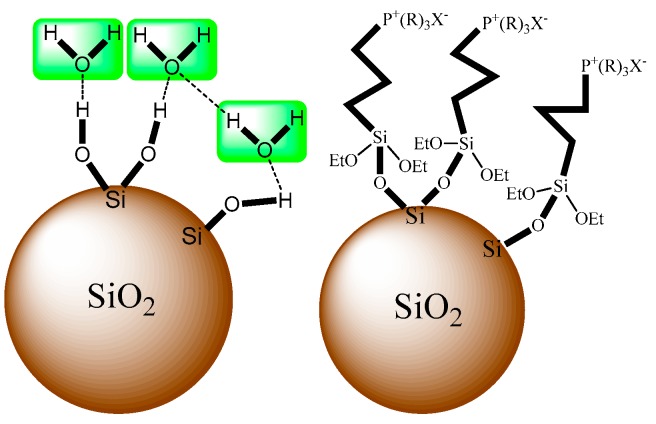
Schematic diagram of the ionic liquid membrane on the surface of the carrier.

**Figure 8 materials-12-01782-f008:**
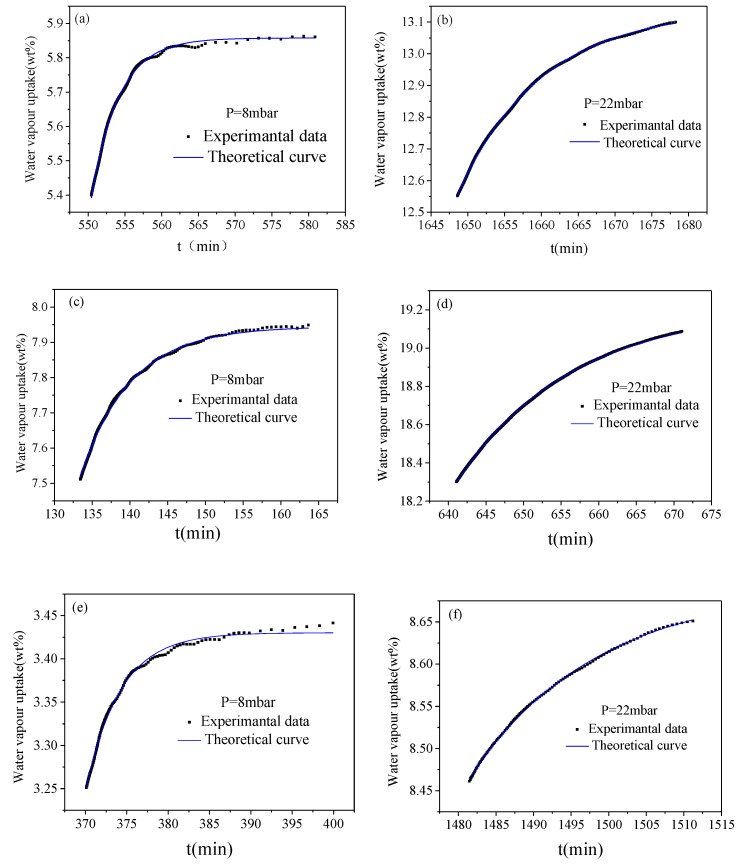
The experimental curves at 25 °C and the theoretical curves, according to Equation (6), for SiO_2_-H (**a**,**b**), SiO_2_-C (**c**,**d**), and Si-P_8_TFSI/SiO_2_-H(I) (**e**,**f**).

**Figure 9 materials-12-01782-f009:**
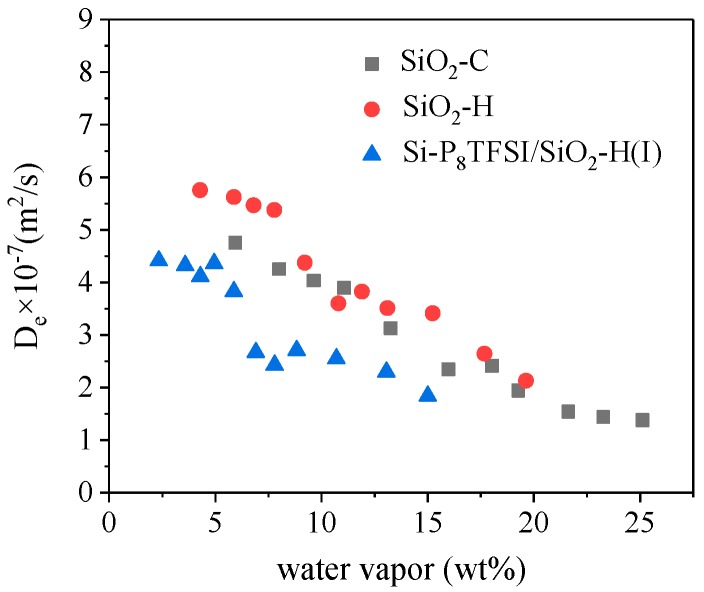
The curve of the effective diffusion coefficients D_e_ versus water vapor uptake.

**Table 1 materials-12-01782-t001:** Silanol group content for bare silica and the total load of phosphonium ILs onto silica surface determined by TGA techniques.

Sample	OH Group Content(mmol/g)	Amount of Grafting (wt%)
SiO_2_-H	3.867	-
SiO_2_-C	5.479	-
Si-P_8_TFSI/SiO_2_-H(I)	-	4.92
Si-P_8_TFSI/SiO_2_-H(II)	-	10.62
Si-P_8_TFSI/SiO_2_-C	-	10.79

**Table 2 materials-12-01782-t002:** Pore structure parameters of samples based on N_2_ sorption isotherms at −196 °C and CO_2_ sorption isotherms at 0 °C.

Sample	N_2_ Sorption Isotherms		CO_2_ Sorption Isotherms		V (CO_2_) (wt%)
	CPV × 10^−1^ (cm^3^·g^−1^)	CSA (m^2^·g^−1^)	CPV × 10^−1^ (cm^3^·g^−1^)	CSA (m^2^·g^−1^)	
SiO_2_-H	4.001	525.3	1.058	306.8	5.51
SiO_2_-C	4.149	635.6	1.241	362.3	6.47
Si-P_8_TFSI/SiO_2-_H(I)	3.303	425.8	0.860	263.2	4.78
Si-P_8_TFSI/SiO_2_-H(II)	3.263	373.1	0.801	238.7	4.27
Si-P_8_TFSI/SiO_2_-C	3.347	455.4	0.980	283.9	4.96

**Table 3 materials-12-01782-t003:** The diffusion coefficient of water vapor sorption in samples at 25 °C.

Pressure (mbar)	De × 10^−7^ (m^2^/s) SiO_2_-C	De × 10^−7^ (m^2^/s) SiO_2_-H	De × 10^−7^ (m^2^/s) Si-P_8_TFSI/SiO_2_vH(I)
5	4.76	5.75	4.41
8	4.25	5.62	4.32
10	4.03	5.46	4.11
12	3.89	5.37	4.36
15	3.12	4.37	3.82
18	2.38	3.61	2.67
20	2.41	3.82	2.43
22	1.94	3.51	2.71
25	1.54	3.42	2.55
28	1.44	2.64	2.98
30	1.38	2.13	1.84

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
