# Peer review of "Investigation on Water Vapor Adsorption of Silica-Phosphonium Ionic Liquids Hybrid Material"

_materials, 2019, doi:10.3390/ma12111782_

Round 1
Reviewer 1 Report
In the reviewed manuscript authors described the results on the investigation on water vapor adsorption of phosphonium ionic liquids immobilized on silica support. The assessment of sorption capacities, especially the elimination of the negative impact of water, is important for the efficiency of CO2 capture. Research has shown that the obtained materials meet such expectations. The work was written carefully and the results obtained were well documented. In my opinion, this manuscript qualifies for publication in Materials, but several issues require explanations or additions. According to the Materials section (item 2.1), the authors also predicted the synthesis of ionic liquids with P444 cation, which was lacking in the discussion of the obtained results. Due to the strongly hydrophobic nature of the NTf2 anion, ionic liquids containing this anion will also have hydrophobic properties. The study does not assess the impact of these properties on CO2 sorption. In general, the authors have not explained the desirability of ILs synthesis with the NTf2 anion, due to the relatively high price of LiNTf2, which is in contradiction with the statement "lower cost in the industry of phosphonium ionic liquids" contained in the Introduction part.
Author Response
Dear editor Kinetics of water adsorption,
We appreciate greatly the reviewers’ effort in reviewing the manuscript and constructive comments and suggestions. Below is our response to the comments.
Reviewer #1:
1. According to the Materials section (item 2.1), the authors also predicted the synthesis of ionic liquids with P444 cation, which was lacking in the discussion of the obtained results.
We deleted the tri-n-butylphosphine(purity:99%) in Materials section (item 2.1) and according to the comment.
2. Due to the strongly hydrophobic nature of the NTf2 anion, ionic liquids containing this anion will also have hydrophobic properties. The study does not assess the impact of these properties on CO2 sorption.
We revised the last paragraph in part 3.2. Water Vapor Adsorption: “The maximum adsorption capacity of water vapor on silica-phosphonium ionic liquid hybrid material is lower than for bare silica with higher pore volume and quantity of hydroxyl group, which is advantageous, as less water vapor will be coadsorbed with CO2 at the high relative humidities that can be expected in post-combustion applications.”
The water hold-up needs to be addressed in adsorption processes under humid conditions, as this can reduce the adsorption capacity for CO2. The focus of this manuscript is to decrease the water vapor adsorption of new porous hybrid material and study the water vapor diffusion at differ relative humidities. Silica grains have been considered to have hydrophilic surfaces and high pore volume, inducing strong adsorption with water molecules. In this work, we obtained new silica-phosphonium ionic liquid hybrid material with better water-resistance and the relationship between the diffusion behaviour and relative humidity. According to the literature, the surface-modifying and regulating the pore size of silica are the effective way to control the amount of adsorbed water vapor in competitive adsorption.
Until now, the mechanism of the different behaviors caused by the presence of water was not quite clear. To research the coadsorption mechanism of CO2 and H2O requires more study on silica surface and porosity which is not the focus of this manuscript, but will be our future work.
3. In general, the authors have not explained the desirability of ILs synthesis with the NTf2 anion, due to the relatively high price of LiNTf2, which is in contradiction with the statement "lower cost in the industry of phosphonium ionic liquids" contained in the Introduction part.
In previous version, this description of property for phosphonium ILs refers to comparing with nitrogen-based ILs. In this version, we revised the description of this point in Introduction part to make it more clearly according to the suggestion., and revised the reference 13.
13. Bradaric, C. J.; Downard, A.; Kennedy, C.; Robertson, A.J; Zhou, Y.H. Industrial preparation of phosphonium ionic liquids. Green Chem, 2003, 5 (2), 143-152.
Reviewer 2 Report
Review attached

Author Response
Dear editor and review,
We appreciate greatly the reviewers’ effort in reviewing the manuscript and their constructive comments and suggestions. Below is our response to their comments.
Reviewer #2:
1.The IR and XPS experiments indicate the ionic liquid was isolated, but its isolation is never described. In the experimental information the authors state that they used a reported procedure (ref. 26). However this reference also refers to a reported procedure which is for a different anion exchange ([BF4]-). There is no way the step
“…the anion was exchanged by reaction of the phosphonium chloride with bistrifluoromethanesulfonamide lithium” can be reproduced with the available information.
We revised the results and discussion in Part 3.1.1 FTIR analysis and described FTIR spectra of the neat phosphonium IL.
In previous version of part 3.1.2, we illustrated the XPS analysis of neat phosphonium IL: “A binding energy of 168.4 eV, 397.4 eV and 688.9 eV is attributed to S 2p, N 1s and F1s respectively in [TFSI] anion [34]. In addition, there is a P 2p peak with a binding energy of 131.0 eV”
Moreover, in this version we revised the 2nd paragraph in part 2.2 Synthesis of grafted silica gel and added the synthesis procedure for anion exchange [TFSI] in detail.
2.The authors state that pore size distribution was calculated by density functional theory. I am almost certain they meant to use a different term here, as there is nothing in the paper that looks even close to a density functional theory calculation. However, assuming the authors did in fact use this technique, then they will need to describe the basis sets, perturbation level, and other parameters that are typically reported with this technique.
We revised the description of the pore size distribution based on density functional theory (DFT) analysis in part 2.3.2. Pore Structure Characteristics: “……by a Quantachrome Autosorb-1-MP, an automatic surface area and pore size analyzer.”
We also added a literature (36) and the description of DFT methods in part of 3.1.2 Pore size distribution: “The DFT model is based on the molecular statistical thermodynamics equation that calculates the specific adsorption amount in an individual pores of a given adsorbate–adsorbent system at a given experimental temperature and pressure by solving the function of grand thermodynamic potential in terms of the distribution of gas density in a specific pore space. ”
36. Landers, J.; Gor, G.Y.; Neimark, A.V. Density functional theory methods for characterization of porous materials. Colloids. Surf. A. Physicochem. Eng. Asp. 2013, 437, 3-32.
3.The experimental section says the water vapor isotherm analyzer was operated at 40 oC, but the abstract, the isotherms in Fig. 6, and the description on page 9, par. 2 all say that the data is for 25 oC.
We revised the temperature in part 2.4. Adsorption Studies and the water vapor isotherm analyzer was operated at 25 oC.
4.The method the authors use to determine the water vapor diffusion coefficients has experimental dependence on both the diffusion coefficient and the material porosity. The porosity is also affected by grafting the IL on silica, as discussed in the last paragraph of section 3.1. How do the authors separate these two contributions? Is porosity (the epsilon parameter in equation 11) assumed or derived? The equations for the fit curves could be given in addition to the figures. This is of central importance to the question the authors are investigating (can CO2 sorption be improved by reducing competition with H2O).
We revised the text in part 3.3 Water diffusion coefficient and added the equation (12) to express the material porosity. We revised the title of the figures 8 to illustrate the equations for the fit curves.
The diffusion in porous solids is more complicated than in liquid. For bare silica and silica-phosphonium ionic liquid hybrid material prepared with grafting method, the particle is considered as a uniform adsorbent. According to Eq. (3), the effective pore diffusivity (De) of porous materials is determined by pore structures which is relation to the porosity. The presence of IL modifies the porous structure of silica gel and pore distribution leading to effect water vapor transport.
On the issue of CO2 and H2O coadsorption, in this version, we revised the last paragraph in part 3.2. Water Vapor Adsorption: “The maximum adsorption capacity of water vapor on silica-phosphonium ionic liquid hybrid material is lower than for bare silica with higher pore volume and quantity of hydroxyl group, which is advantageous, as less water vapor will be coadsorbed with CO2 at the high relative humidities that can be expected in post-combustion applications.”
The water hold-up needs to be addressed in adsorption processes under humid conditions, as this can reduce the adsorption capacity for CO2. The focus of this manuscript is to decrease the water vapor adsorption of new porous hybrid material and study the water vapor diffusion model. In this work, we obtained new silica-phosphonium ionic liquid hybrid material with better water-resistance and the relationship between the diffusion behaviour and relative humidity. According to the literature, the surface-modifying and regulating the pore size of silica are the effective way to control the amount of adsorbed water vapor in competitive adsorption of H2O and CO2. Until now, the mechanism of the different behaviors caused by the presence of water was not quite clear. To research the coadsorption mechanism of CO2 and H2O requires more study on silica surface and porosity and will be our future work.
5. The introduction is reasonably well referenced (supported ionic liquids and phosphonium ionic liquids are very well explored systems), but throughout the discussion there is little comparison to the literature. The only references used are those explaining where the techniques came from. Ultimately this paper should say something about how the IL treated silica compares to known materials. For instance, are the diffusion coefficients of water vapor similar to silanized silica?
We added the references in part 3.3 Water diffusion coefficient and the diffusion coefficients of water vapor in this work are close to those reported silica [41-43]
[41] Oehler, A.; Tomozawa, M. Water diffusion into silica glass at a low temperature under high water vapor pressure. J. Non-Cryst. Solids. 2004, 347(1-3), 211-219.
[42] Aristov, Y.I.; Tokarev, M.M.; Freni, A.; Glaznev, I.S.; Restuccia, G. Kinetics of water adsorption on silica Fuji Davison RD. Micropor. Mesopor. Mater. 2006, 96 (1), 65-71.
6.Phosphonium ionic liquids have been well explored and even commercially applied as stationary phases for gas chromatography. Has any of the data exploring this application produced any usable diffusion coefficients?
Given the properties of phosphonium ILs, they are evaluated and employed as stationary phases either commercial or laboratory made for gas chromatography. The main research is mainly focused on the column efficiency for alkanes, polycyclic aromatic hydrocarbons, aromatic isomers, alcohols, and chlorinated benzenes compounds, and so on. The IL stationary phase also exhibited good selectivity and high separation efficiency. The IL column was usually prepared by the static coating method.
The gas diffusivity for silica-phosphonium ionic liquid hybrid material with various pore structures is different from the bulk ILs. Therefore, we do not add extra discussion on this point.
Round 2
Reviewer 1 Report
I accept all comments and responses. In the current version, the manuscript qualifies for publication in Materials.
Author Response
Dear editor and reviewer,
We appreciate greatly the reviewer’s effort in reviewing the manuscript.
With best wishes!
Jiamei Zhu
(Correspondence author)
Reviewer 2 Report
The authors have addressed most of my methodological concerns although I recommend the inclusion of a few specific details as stated below. Although the conclusions of the study do not suggest the material is particularly promising, I believe this article can contribute to discussions on method development for this potentially important yet under-investigated phenomenon.
For lines 94-97, include how the LiCl which must have been produced as a by-product of this reaction was separated (filtering, extraction, etc.).
The authors added a reference explaining the DFT method, but in the experimental (lines 133-134) they need to explain how this method was actually implemented (Gaussian, GAMESS, etc.).
Throughout the manuscript there are many minor English problems such as verb conjugation, use of plurals, and word endings.
Author Response
Dear editor and reviewer,
We appreciate greatly the reviewer’s effort in reviewing the manuscript and constructive comments and suggestions. Below is our response to the comments.
Reviewer #2:
1. For lines 94-97, include how the LiCl which must have been produced as a by-product of this reaction was separated (filtering, extraction, etc.).
We revised the text (blue font) in lines 94-97.
2. The authors added a reference explaining the DFT method, but in the experimental (lines 133-134) they need to explain how this method was actually implemented (Gaussian, GAMESS, etc.).
We revised the text (blue font) in the experimental (lines 137-138).
3. Throughout the manuscript there are many minor English problems such as verb conjugation, use of plurals, and word endings.
We revised the grammar of the manuscript.
With best wishes!
Jiamei Zhu
(Correspondence author)